# Decompression with or without Fusion for Lumbar Synovial Cysts—A Systematic Review and Meta-Analysis

**DOI:** 10.3390/jcm12072664

**Published:** 2023-04-03

**Authors:** Alberto Benato, Grazia Menna, Alessandro Rapisarda, Filippo Maria Polli, Manuela D’Ercole, Alessandro Izzo, Quintino Giorgio D’Alessandris, Nicola Montano

**Affiliations:** Department of Neuroscience, Neurosurgery Section, Fondazione Policlinico Universitario Agostino Gemelli IRCCS—Università Cattolica del Sacro Cuore, 00168 Rome, Italy

**Keywords:** lumbar synovial cyst, lumbar posterior decompression, lumbar decompression and fusion

## Abstract

The management of symptomatic lumbar synovial cysts (LSC) is still a matter of debate. Previous systematic reviews did not stratify data according to different treatment techniques or incompletely reported comparative data on patients treated with lumbar posterior decompression (LPD) and lumbar decompression and fusion (LDF). The aim of our study was to compare LPD and LDF via a systematic review and meta-analysis of the existing literature. The design of this study was in accordance with the 2020 Preferred Reporting Items for Systematic Reviews and Meta-Analyses (PRISMA) guidelines. The review questions were as follows: among patients suffering from symptomatic lumbar synovial cysts (population) and treated with either posterior lumbar decompression or posterior decompression with fusion (intervention), who gets the best results (outcome), in terms of cyst recurrence, reoperation rates, and improvement of postoperative symptoms (comparison)? The search of the literature yielded a total of 1218 results. Duplicate records were then removed (*n* = 589). A total of 598 articles were screened, and 587 records were excluded via title and abstract screening; 11 studies were found to be relevant to our research question and were assessed for eligibility. Upon full-text review, 5 were excluded because they failed to report any parameter separately for both LPD and LDF. Finally, 6 studies for a total of 657 patients meeting the criteria stated above were included in the present investigation. Our analysis showed that LDF is associated with better results in terms of lower postoperative back pain and cyst recurrence compared with LPD. No differences were found in reoperation rates and complication rates between the two techniques. The impact of minimally invasive decompression techniques on the different outcomes in LSC should be assessed in the future and compared with instrumentation techniques.

## 1. Introduction

Spinal juxtafacet cysts are a relatively uncommon finding; they are thought to be an expression of degenerative processes related to joint micro-instability [1,2,3,4,5]. Despite their rarity, it is important to understand what they are, their causes, and how they are treated. Most of them are found in the lumbar spine (LS) and, when symptomatic, are associated with low back pain and radiculopathy [3,5,6].

They are typically seen as a calcified cystic lesion adjacent to a facet joint. CT may also show adjacent facet joint arthropathy and/or the presence of gas. This entity cannot be reliably distinguished from ganglion cysts on standard MRI. However, communication with the joint space after intra-articular injection with contrast reliably differentiates between them. The presence of gas within the cyst is pathognomonic for a synovial cyst. Facet joint cysts may contain complex fluid because of internal debris or hemorrhage. Neural-based cysts can usually be differentiated by imaging as these cysts show intimate relation with the adjacent nerve, rather than with the adjacent joint space. Calcification within the cyst wall appears as low signal intensity on both T1 and T2 weighted images whereas hemorrhagic cysts display increased intensity compared to CSF likely due to T1 shortening from methemoglobin. The cysts do not always possess the signal characteristics of a simple cyst, so contrast administration may be needed in some cases. The definitive diagnosis to distinguish between the two types of cysts may only be made via pathologic examination. While synovial cysts have a lining of cuboid, epithelium-like synovial cells with a clear or xanthochromic fluid, ganglion cysts have a gelatinous, proteinaceous material with myxoid degeneration of the fibrous adventitial tissue, but importantly, no synovial lining. However, distinguishing between ganglion and synovial cysts is only of histologic value, as their clinical features, treatment, and prognosis are the same.

It is crucial to understand the symptoms and risk factors associated with lumbar synovial cysts (LSCs) to provide the best possible care for patients who suffer from this condition. Two-thirds of these cysts are located at the L4-L5 level, which is the most mobile lumbar level [6,7,8]. The optimal management of symptomatic LSCs is still a matter of debate. While there are many different approaches to treating LSCs, the most widely adopted treatment approach is posterior decompression and cyst resection alone. This approach has been shown to have good outcomes in the literature [3,7]. Due to the mentioned association between LSC and spinal micro-instability, lumbar decompression alone has been associated with a risk of cyst recurrence, recurrence of symptoms (especially axial back pain), and establishment of macro-instability, with the need for revision surgery [7,8]. To avoid these risks, additional lumbar fusion is sometimes performed as part of the same procedure. However, this is only necessary in 1–5% of cases [6,7]. Although past publications have suggested that lumbar decompression and fusion (LDF) could outperform lumbar posterior decompression (LPD) [8,9], this problem has never been assessed systematically before. This is an important gap in our understanding of LSC treatment. Previous systematic reviews did not stratify data according to different treatment techniques [3] or incompletely reported comparative data on patients treated with LDF [6,7]. Our study aims to fill this gap by comparing LPD and LDF via a systematic review and meta-analysis of the existing literature. By doing so, we hope to provide a clearer understanding of the relative benefits and drawbacks of these two treatment approaches.

## 2. Materials and Methods

### 2.1. Study Design and Review Question

The study design followed the 2020 Preferred Reporting Items for Systematic Reviews and Meta-Analyses (PRISMA) guidelines. The review questions were formulated using the population, intervention, comparison, and outcome (PICO) scheme as follows: among patients with symptomatic lumbar synovial cysts (population) treated with either posterior lumbar decompression or posterior decompression with fusion (intervention), who achieved the best results (outcome) in terms of cyst recurrence, reoperation rates, and improvement of postoperative symptoms (comparison)?

### 2.2. Inclusion Criteria and Outcome Measurement

Different medical databases (PubMed and Scopus) were screened for eligible scientific reports. The keywords “lumbar”, “synovial”, and “cyst” (MeSH) were used in any possible combination. The last search was conducted in January 2022. To identify relevant reports, two reviewers (A.B. and G.M.) independently screened the abstracts and reference lists. Any differences were resolved by consensus with a third senior author (N.M.).

### 2.3. Summary of the Included Studies

Studies Were Included if They Met the following Criteria
Comparative, prospective, or retrospective studies in English on LPD vs. LDF in patients with symptomatic lumbar synovial cysts;Reporting at least one outcome measurement among recurrence rates, reoperation rates, postoperative clinical picture, and/or postoperative complications for each of the two treatment modalities;Used a study population of more than 10 patients.

The following demographic and clinical outcome variables were collected for meta-analysis purposes:-Demographic data (age, sex);-Preoperative clinical data (back pain, radiculopathy, motor deficit, sensory deficit, bowel/bladder deficit, preoperative spondylolisthesis);-Postoperative outcome data (synovial cyst recurrence, reoperation, postoperative back pain, postoperative radiculopathy, postoperative infection, CSF leakage, and postoperative length of stay).

### 2.4. Statistical Analysis

Statistical analyses were performed using Review Manager (RevMan) [Version 5.4, The Cochrane Collaboration, 2020] and applying the random-effect model. Heterogeneity was tested using the chi-square test and quantified by calculating the I^2^ statistic, in which *p* < 0.05 and I^2^ > 50% were considered statistically significant. For the pooled effects, the odds ratio (OR) was calculated for the dichotomous variables. Continuous variables are presented as mean differences and 95% confidence intervals (CI), whereas dichotomous variables are presented as ORs and 95% CI. Publication bias was tested using funnel plots.

## 3. Results

The search of the literature yielded a total of 1218 results. Duplicate records were then removed (*n* = 589). A total of 598 articles were screened, and 587 records were excluded via title and abstract screening; 11 studies were found to be relevant to our research question and were assessed for eligibility (Figure 1). Upon full-text review, 5 were excluded because they failed to report any parameter separately for both LPD and LDF. Finally, 6 studies for a total of 657 patients meeting the criteria stated above were included for the present investigation (Table 1). The meta-analysis results are summarized in Table 2. The forest plots and funnel plots for each investigated variable are reported in the Appendix A.

### 3.1. Pre-Operative Clinical Data

The data on preoperative symptoms were reported in two papers. Among them, radiculopathy was more frequently observed in those patients who were treated with LPD (OR 5.12 [95% CI 1.5–17.46], *p* = 0.009) (I^2^ = 0%, *p =* 0.39); the analysis of other preoperative clinical data (motor deficit, sensory deficit, back pain, bowel/bladder dysfunction) did not show significant differences between the two groups. Preoperative rates of spondylolisthesis were also not found to differ significantly between treatment groups (Table 2 and Appendix A).

### 3.2. Outcome Factors

Among the available outcome factors, cyst recurrence (OR 11.08 [95% CI 2.10–58.61], *p* = 0.005) (I^2^ = 0%, *p* = 0.97) and postoperative back pain (OR 2.78 [95% CI 1.46–5.30], *p* = 0.002) (I^2^ = 6%, *p* = 0.30) were assessed in three and two of the six papers included, respectively, and appeared significantly more frequently in those treated with LPD alone. Hospital length-of-stay, assessed in two papers, was significantly longer in patients treated with LDF, with a mean difference of 2.87 days (95% CI 2.10–3.65, *p* < 0.00001). Reoperation rates, the most consistently reported parameter (four papers), did not differ significantly between the two groups. In the same way, the remaining evaluated factors (postoperative radiculopathy, infections, and CSF leakage, each assessed in two papers) did not appear significantly different between the two treatment groups (*p* > 0.05) (Table 2 and Appendix A).

### 3.3. Publication Bias

In the meta-analysis conducted, it was observed that the funnel plots were symmetrical with respect to the different variables studied. This indicates that publication bias did not appear to have a significant impact on the results. It is worth noting that the data collected from each study were carefully evaluated and included in the analysis to ensure a complete and accurate picture. Appendix A provides additional details on the analysis conducted as well as the variables used in the study.

## 4. Discussion

LSCs are a relatively uncommon occurrence, being reported in only about 0.6% of lumbar scans performed for various reasons [14,15,16,17]. These cysts are believed to be an expression of facet joint degeneration brought on by spinal micro-instability, which leads to excessive deposition of extracellular matrix and extrusion of synovia through the weakened joint capsule [1,15,16,18,19,20]. Although some authors have proposed a broader definition of “cystic formation of the mobile spine” due to the heterogeneity in their morphologic and anatomical features, this definition has little clinical relevance [16,17]. Despite the rarity of LSCs, they are of great interest to medical professionals due to their potential to cause significant discomfort and pain to those affected.

The vast majority of LSCs occur in the lumbar spine, with two-thirds of cases being located at the L4-L5 level, the spinal segment where maximum loading and mobility intersect [1,3,5,6,8,21]. This is seen as indirect evidence of the link between LSCs and instability [1]. Furthermore, previous studies have directly linked the occurrence of LSCs to the presence and severity of lumbar facet joint osteoarthritis [21,22] and lumbar spondylolisthesis [1,21,22,23,24], which is present as an associated condition in about 30–40% of cases [1,3]. Putative associations with a traumatic etiology have been discarded with the accumulation of cases in the literature [6].

The most common symptomatic repercussion of LSCs is radiculopathy, which occurs in 92% of cases [6]. This is caused by direct compression of nerve roots [25,26]. In rare cases, an onset of severe symptoms has been reported after intracystic hemorrhage [27,28]. About 70% of patients also complain of low back pain which can be related to abnormal strain of lumbar osteoligamentous structures due to the phenomena described above [29].

To date, over 800 reports of variously treated patients have been published [6,9]. Among these, about two-thirds were treated with cyst de-compression via LPD (i.e., laminectomy, hemilaminectomy, interlaminar approach, or flavectomy); percutaneous cyst aspiration comprises most remaining reports. Despite the widely echoed concerns in the literature about instability, only a small percentage of patients (about 1–5%) were managed with additional lumbar fixation [6,9]. Previous systematic reviews have found high rates of symptomatic relief (90%), low rates of reoperation, and satisfactory long-term outcomes after simple LPD for LSCs [6], with low operative complication rates. However, studies with longer follow-up periods have highlighted non-negligible rates of recurrent back pain [3] after simple LPD. For these reasons, the best surgical treatment for LSCs is still a matter of debate [8,9,30,31].

In our meta-analysis we found that LDF is associated with lower rates of postoperative low back pain and symptomatic cyst recurrence. This can be easily explained by the fact that the stabilization of the involved segments could alleviate the stress responsible for axial pain and block the process responsible for cyst formation. Selection bias seems to not play a role between the two treatment groups. Interestingly, differences in preoperative rates of spondylolisthesis did not reach statistical significance between the two groups suggesting that this factor did not play a significant role in deciding which technique to use in the treatment of LSCs. The fact that a pre-operative radiculopathy is more frequently reported in LPD group confirms the fact that when approaching a LSC the attention is focused more on the nerve decompression than on the overall spine function. It is important to underline that not all LPD techniques are the same, and over the years various less disruptive decompression modalities have been developed. Unfortunately, the data reported in the included papers did not allow for separate analyses distinguishing the LPD techniques adopted. For example, in the paper by Xu and colleagues, 43% of LPD patients were treated with laminectomy and 57% with hemilaminectomy [8]; Page and colleagues reported decompression techniques ranging from laminotomy to multilevel laminectomy without providing the number of patients treated with each modality [9]. It could be hypothesized that less invasive techniques, such as cyst decompression via an interlaminar approach or flavectomy might be associated with less postoperative instability [10,32,33]. Unfortunately, no comparative series comparing these minimally invasive approaches with fusion strategies are available in the literature. Nonetheless, significant rates of symptomatic cyst recurrence on adjacent facets have been reported even in studies focused on minimally disruptive LPD approaches (interlaminar or hemilaminectomy) [12,13], documenting a trend towards propagation of micro-instability regardless of the entity of iatrogenic posterior band disruption.

In addition, no comparisons could be made among fusion techniques (instrumented vs. non-instrumented), as not all studies explicitly discriminated between techniques [9,11,34]. In included two studies [35,36], only non-instrumented fusions were performed, while in the other studies, variable combinations of both instrumented and non-instrumented fusion were used. However, the only outcome parameters that could be analyzed in relation to these studies were reoperation rate and preoperative spondylolisthesis, respectively. It has been shown that, in the lumbar spine, instrumented fusion grants higher fusion rates and lower incidences of pseudoarthrosis with respect to non-instrumented fusion [34]. Separate outcome parameters for instrumented versus non-instrumented fusion were clearly stated only in one of the included studies, where instrumented fusion appears to be associated with even lower rates of postoperative back pain and cyst recurrence [8]. These data are anyway insufficient to formulate any recommendation on the preferred fusion technique. Therefore, it is vital to conduct further research to compare the different LPD techniques and fusion strategies. This will enable us to identify the most effective and safe method to treat LSCs. It is also important to consider the potential risks and benefits of each technique when deciding which method to use. By doing so, we can ensure that the best possible outcomes are achieved for patients with LSCs.

The addition of fusion to lumbar decompression requires a more extensive surgical approach, which can lead to longer operative times and potentially increase the risk of complications. Despite this theoretical concern, our analysis did not reveal any significant differences in the rates of complications, such as infection and cerebrospinal fluid leak, between the treatment groups. However, it is worth noting that only two papers provided detailed data on postoperative complications for each treatment, which limited the number of patients available for comparison.

In addition to the potential risks associated with fusion, it is also important to consider the impact on hospital stay. Our analysis found that fusion techniques generally resulted in longer hospital stays compared to lumbar decompression alone. Specifically, patients undergoing LDF had a mean increase of 2.87 days in their hospital stay. These findings suggest that while fusion may offer certain benefits, such as improved long-term outcomes, it is important to carefully weigh the potential risks and benefits before deciding on the best course of treatment for each individual patient.

### Limitations

The main limitations of our meta-analysis include the small number of studies that met the inclusion criteria, the retrospective nature of all the included studies, and the different techniques used for both LPD and LDF among the included studies. However, it is worth noting that our findings suggest that LDF may be associated with better outcomes in terms of lower postoperative back pain and cyst recurrence compared with LPD. To further investigate this potential association, future studies could focus on conducting prospective trials comparing LPD and LDF. Additionally, researchers could explore the use of standardized techniques for both LPD and LDF to reduce the variability between studies. Overall, while there are several variables that may make it challenging to confirm the association between LDF and improved outcomes, further research in this area could provide more insight into the potential benefits of LDF as a treatment option for patients.

## 5. Conclusions

Our analysis indicates that LDF may be a more effective treatment option for reducing post-operative back pain and cyst recurrence, according to the available comparative studies in the literature. Although these studies are all retrospective, they suggest that LDF may have advantages over other LPD techniques. However, it is important to note that reoperation rates and complication rates do not differ significantly between LDF and LPD techniques. Despite these findings, it is crucial to acknowledge that the current literature is limited as it lacks prospective, well-designed, comparative studies that can comprehensively compare LPD and LDF in the treatment of LSCs. Further research is necessary in this area, especially in assessing the impact of minimally invasive decompression techniques on different outcomes in LSCs. Additionally, future studies should also compare the effectiveness of the different instrumentation techniques. Considering the limited research in this area, it is important to remain cautious when interpreting the results of the available studies.

In summary, while the available literature suggests that LDF may be a more effective treatment option for reducing post-operative back pain and cyst recurrence compared to LPD, further research is necessary before drawing definitive conclusions.

## Figures and Tables

**Figure 1 jcm-12-02664-f001:**
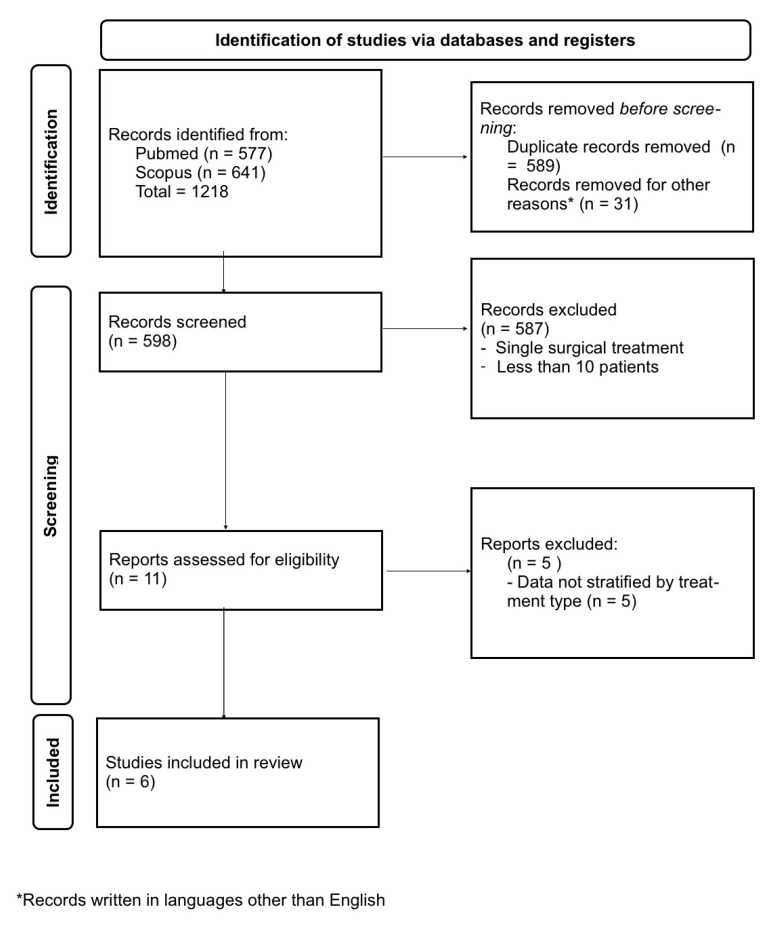
Analysis according to the PRISMA guidelines.

**Table 1 jcm-12-02664-t001:** 1 Summary of the studies comparing posterior lumbar decompression with lumbar decompression and fixation for the treatment of lumbar synovial cysts included in the meta-analysis. (Y = years; MOS = months after surgery).

Study	Decompression (No. of Patients)	Fusion (No. of Patients)	Mean Age (Y)	Follow-Up (MOS.)
**Page et al., 2020 [9]**	88	39	63.2	-
**Wun et al., 2019 [10]**	55	32	63.2 ± 8.8	65.1 ± 48.6
**Xu et al., 2010 [8]**	90	74	63.00 ± 12.04	16.5 ± 9.2
**Weiner et al., 2007 [11]**	23	23	73	116
**Khan et al., 2005 [12]**	13	26	63.3	26
**Lyons et al., 2000 [13]**	176	18	66	26 (76% of cohort)

**Table 2 jcm-12-02664-t002:** Summary of the results of the meta-analysis. (OR = Odds Ration; CI = Confidence interval).

FACTORS	OR	95% CI	*p* Value	I^2^	I^2^ *p* Value
**Sex (Male)**	1.68	0.61–4.65	0.32	69%	0.04
**Back pain**	1.28	0.71–2.33	0.41	17%	0.27
**Radiculopathy**	**5.12**	**1.5–17.46**	**0.009**	**0**	**0.39**
**Motor deficit**	3.05	0.13–73.89	0.49	89%	0.003
**Sensory deficit**	2.17	0.13–37.2	0.59	95%	<0.00001
**Bowel/bladder deficit**	0.28	0.02–3.67	0.33	46%	0.17
**Pre-op spondylolysthesis**	0.22	0.04–1.15	0.07	71%	0.06
**Cyst recurrence**	**11.08**	**2.10–58.61**	**0.005**	**0**	**0.97**
**Reoperation**	1.89	0.36–10.03	0.46	68%	0.03
**Post-op back pain**	**2.78**	**1.46–5.30**	**0.002**	**6%**	**0.30**
**Post-op radiculopathy**	1.29	0.64–2.58	0.47	0	0.73
**Infections**	0.16	0.02–1.36	0.09	0	0.53
**CSF leakage**	0.47	0.05–4.18	0.49	0	0.65
**Length of stay**	**−2.87 (WMD)**	**−3.65–2.10**	**<0.00001**	**0**	**0.91**

OR: odds ratio; CI: confidence interval; WMD: weight mean difference.

## Data Availability

Not applicable.

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
