# Peer review of "Decompression with or without Fusion for Lumbar Synovial Cysts—A Systematic Review and Meta-Analysis"

_jcm, 2023, doi:10.3390/jcm12072664_

Round 1

Reviewer 1 Report

The authors review the literature to compare decompression vs decompression and fusion for the treatment of synovial cysts. They find that fusion patients have less postoperative back pain and cyst recurrence compared to decompression only. The supporting data are not particularly strong, but the information is somewhat useful in future surgical decision making.

Author Response

REVIEWER 1

The authors review the literature to compare decompression vs decompression and fusion for the treatment of synovial cysts. They find that fusion patients have less postoperative back pain and cyst recurrence compared to decompression only. The supporting data are not particularly strong, but the information is somewhat useful in future surgical decision making.

Dear Reviewer, Thank you for taking the time to review our manuscript. We appreciate your valuable feedback.

REVIEWER 2

Thank you for your retrospective meta analysis on surgery for lumbar facet cysts.  The question as to whether or not to fuse is often the question when treating these patients and your article is an interesting read.  I agree with your assessment that newer less invasive decompression alone techniques are being used now and makes looking at the older articles doing full laminectomies hard to compare.   Similarly, for the fusion groups non instrumented fusions is rare and makes present day comparisons hard.  There is also the question of brace therapy options after cyst resection alone as way to prevent recurrance....  all in all there are many variables that make it hard to confirm that LDF is associated with better results in terms of lower postoperative back pain and cyst recurrence compared with LPD. Interestingly there was no differences found in 244 reoperation rates suggesting that cyst recurrence in the laminectomy alone group is well tolerated given the decompression and may support it as a primary treatment option given the added expense, operative time, and hospital stay in fusion patients.

Dear Reviewer, Thank you for taking the time to review our manuscript. We appreciate your valuable feedback.

REVIEWER 3

I was pleased to read this interesting article that reports- Decompression with or without fusion for lumbar synovial 2 cysts – a systematic review and meta-analysis. I appreciate going through your article. It need major revision. Please go through the comments.

Dear Reviewer,

Thank you for taking the time to review our manuscript. We appreciate your valuable feedback and suggestions, which have helped us to improve the quality of our work. We have carefully considered all of your comments and have made the necessary revisions to address the points you  raised. Below, you will find a point-to-point response to your comments. We believe that these changes have significantly strengthened the manuscript and have addressed the concerns raised.

Abstract:

Reoperation rates, and postoperative symptoms (comparison)? 598 non-duplicated articles were reviewed. What does this means? Please correct the sentence and grammar. Need extensive English style and grammar.

The review question was formulated using the population, intervention, comparison, and outcome (PICO) scheme as follows: among patients with symptomatic lumbar synovial cysts (population) treated with either posterior lumbar decompression or posterior decompression with fusion (intervention), who achieved the best results (outcome) in terms of cyst recurrence, reoperation rates, and improvement of postoperative symptoms (comparison)? To improve clarity, the sentence was rephrased as suggested in both abstract and methods section. Specifically, we had a mother tongue English speaker review the manuscript, and we have rephrased the sentence in question to address any concerns about language or clarity.

Among those, 589 were excluded because they did not include both 20 LPD and LDF. Among the remaining 11 studies, 5 were excluded because they failed to report any 21 parameter separately for both LPD and LDF. Finally, 6 studies for a total of 657 patients meeting the 22 criteria stated above, were included in the present investigation. Please check calculation totally wrong. That means all the authors have not read the article.

We carefully reviewed our calculation and rephrased the sentence as follows: “The search of the literature yielded a total of 1218 results. Duplicate records were then removed (n = 589). A total of 598 articles were screened, and 587 records were excluded via title and abstract screening; 11 studies were found to be relevant to our research question and were assessed for eligibility. Upon full-text review, 5 were excluded because they failed to report any parameter separately for both LPD and LDF. Finally, 6 studies for a total of 657 patients meeting the criteria stated above were included for the present investigation”

Material & Methods: PRISMA guidelines. Have you registered your study under PICO or PROSPERO. Please write the registration number.

We have not registered our study yet. We will update the registration number as soon as the required

Result: 598 non-duplicated articles were reviewed. Among those, 589 were excluded be- 97 because they did not include both LPD and LDF. Among the remaining 11 studies, 5 98 were excluded because they failed to report any parameter separately for both LPD 99 and LDF. Finally, 6 studies for a total of 657 patients meeting the criteria stated 100 above were included in the present investigation- Please check it again.

We carefully reviewed our calculation and rephrased the sentence as follows: “The search of the literature yielded a total of 1218 results. Duplicate records were then removed (n = 589). A total of 598 articles were screened, and 587 records were excluded via title and abstract screening; 11 studies were found to be relevant to our research question and were assessed for eligibility (Figure 1). Upon full-text review, 5 were excluded because they failed to report any parameter separately for both LPD and LDF. Finally, 6 studies for a total of 657 patients meeting the criteria stated above were included for the present investigation (Table 1).”

Figure 1. Analysis according to the PRISMA statement- Please check and correct the numbers of studies included and excluded.

See above.

What is the originality and novelty of the review article? 

Our analysis focused on a comparison of Laminectomy with fusion (LDF) versus lumbar posterior decompression (LPD) for the management of symptomatic lumbar synovial cysts (LSC). In the literature, good outcomes are generally reported with posterior decompression and cyst resection alone, which is the most widely adopted treatment approach. Although past publications have suggested that LDF could outperform LPD, this problem has never been assessed systematically before. Previous systematic reviews did not stratify data according to different treatment techniques or incompletely reported comparative data on patients treated with LDF.

References: Please correct reference- Ramhmdani S, Ishida W, Perdomo-Pantoja A, Witham TF, Lo S-FL, Bydon A (2019) Synovial Cyst as a Marker for Lumbar Insta- 355 bility: A Systematic Review and Meta-Analysis. World Neurosurgery 122:e1059–e1068. doi: 10.1016/j.wneu.2018.10.228. 1 A is written before year. Bold the Year of publication. Follow the MDPI JCM guidelines for referencing.

We corrected the references as required.

Reviewer 2 Report

Thank you for your retrospective meta analysis on surgery for lumbar facet cysts.  The question as to whether or not to fuse is often the question when treating these patients and your article is an interesting read.  I agree with your assessment that newer less invasive decompression alone techniques are being used now and makes looking at the older articles doing full laminectomies hard to compare.   Similarly, for the fusion groups non instrumented fusions is rare and makes present day comparisons hard.  There is also the question of brace therapy options after cyst resection alone as way to prevent recurrance....  all in all there are many variables that make it hard to confirm that LDF is associated with better results in terms of lower postoperative back pain and cyst recurrence compared with LPD.

Interestingly there was no differences found in 244 reoperation rates suggesting that cyst recurrence in the laminectomy alone group is well tolerated given the decompression and may support it as a primary treatment option given the added expense, operative time, and hospital stay in fusion patients.

Author Response

(The authors gave the same response as above.)

Reviewer 3 Report

Dear authers it is good systamatic review article, but need many corrections. Please follow the comment file attached. 

Best regards.

Author Response

(The authors gave the same response as above.)

Round 2

Reviewer 3 Report

Dear authors it is fine now. Best of luck